# Exploring the Immune Response against RSV and SARS-CoV-2 Infection in Children

**DOI:** 10.3390/biology12091223

**Published:** 2023-09-09

**Authors:** Rafaela Pires da Silva, Bibiana Liberman Thomé, Ana Paula Duarte da Souza

**Affiliations:** Laboratory of Clinical and Experimental Immunology, Pontifical Catholic University of Rio Grande do Sul, Porto Alegre 90619-900, Brazil; rafaela.s@edu.pucrs.br (R.P.d.S.); bibiana.thome@edu.pucrs.br (B.L.T.)

**Keywords:** RSV, SARS-CoV-2, immune response, child, lung, infants

## Abstract

**Simple Summary:**

Respiratory viral infections are among the most common diseases that affect children. Respiratory syncytial virus (RSV) is the most common pathogen causing lower tract infections in children and is responsible for a significant number of hospital admissions, whereas, in adults, its clinical importance is less significant. In contrast, SARS-CoV-2, the virus responsible for the COVID-19 pandemic, rarely causes severe disease in children. In this review, we discuss the differences in susceptibility between these two viruses in children with a focus on the immune response. A better understanding of the immune responses induced by SARS-CoV-2 and RSV, as well as their unique roles in viral evasion, is essential for the development of vaccines and new drugs.

**Abstract:**

Viral respiratory tract infections are a significant public health concern, particularly in children. RSV is a prominent cause of lower respiratory tract infections among infants, whereas SARS-CoV-2 has caused a global pandemic with lower overall severity in children than in adults. In this review, we aimed to compare the innate and adaptive immune responses induced by RSV and SARS-CoV-2 to better understand differences in the pathogenesis of infection. Some studies have demonstrated that children present a more robust immune response against SARS-CoV-2 than adults; however, this response is dissimilar to that of RSV. Each virus has a distinctive mechanism to escape the immune response. Understanding the mechanisms underlying these differences is crucial for developing effective treatments and improving the management of pediatric respiratory infections.

## 1. Introduction

Ranking among the most prevalent illnesses, viral respiratory infections in the upper and lower respiratory tract pose a persistent public health challenge due to their substantial impact on morbidity and mortality worldwide, especially at a young age [1,2]. Notably, children and infants are particularly vulnerable to respiratory infections, often enduring five to six episodes annually [3,4].

Different viruses can cause respiratory infections, including rhinoviruses, enteroviruses, adenoviruses, parainfluenza, influenza, human metapneumovirus, respiratory syncytial virus (RSV), coronaviruses, and the current severe acute respiratory syndrome coronavirus 2 (SARS-CoV-2) [5,6]. RSV is recognized as a major cause of lower respiratory tract infection (LRTI) among children, especially the younger ones [7,8], responsible for 13–22% of deaths from LRTI in children, and resulting in over 100,000 global deaths annually [9]. RSV infection encompasses a range of symptoms, from mild to lethal illnesses, such as mild rhinitis, bronchiolitis, and pneumonia [7]. Classic clinical signs of bronchiolitis caused by RSV begin with symptoms typical of an upper respiratory tract viral infection, such as nasal congestion, progressing to the lower respiratory tract within a few days [10].

The SARS-CoV-2 virus, responsible for the coronavirus disease (COVID-19), has had a great impact worldwide since the outbreak of the pandemic in 2019, resulting in over 653 million cases worldwide and a total of 6 million deaths. However, only 0.4% of these deaths were in children [11,12]. Studies also suggest differences between age groups, with children between 0 and 9 years having an approximate rate of 0.18 deaths per 100,000 and children between 10 and 18 years having a rate of 0.37 per 100,000 [12,13,14]. In comparison to adults and differing from other respiratory viruses, most pediatric cases of COVID-19 are asymptomatic or mild; among the symptoms of COVID-19 in children are fever, cough, and nasal congestion [15]. With the emergence of the Delta and Omicron variants in 2021, COVID-19 pediatric cases increased, but not mortality rates [16]. In rare cases, children who contract SARS-CoV-2 might experience a late manifestation of the disease, multisystem inflammatory syndrome (MIS-C) [17], which is identified mainly by fever, rash, and gastrointestinal symptoms. MIS-C usually affects children between 8 and 9 years old and can be severe; however, it is a treatable condition, and most children recover fully. MIS-C occurs in approximately 3 of 10,000 infected persons under 21 years of age, with a mortality rate of 0.8% [18].

The COVID-19 pandemic has had a huge impact on RSV circulation, with a surge of RSV cases off-season [19]. This phenomenon was associated with the use of non-pharmaceutical interventions introduced during the pandemic [20]. The proportion of co-infections with RSV and SARS-CoV-2 in children is usually low but with a severe clinical phenotype [21]. However, in adults, co-infection with SARS-CoV-2 and RSV does not lead to more severe cases, in contrast to influenza together with SARS-CoV-2 which is associated with an increased need for invasive mechanical ventilation [22].

The difference in susceptibility between children infected with RSV and those infected with SARS-CoV-2 is notable, considering that children infected with RSV develop severe disease, whereas in cases of SARS-CoV-2 infection in children, the disease tends to be less severe than in adults. Understanding why these two highly aggressive agents produce different responses in children is essential for future intervention.

### RSV, SARS-CoV-2, and Cellular Airway Infections

Both viruses are enveloped; however, their genomes have different sizes and polarities. RSV has 11 proteins encoded in a genome of 15 kb of negative-sense single-stranded RNA [23]. SARS-CoV-2 has a genome of 30 kb, composed of a positive-sense single-stranded RNA that encodes 29 proteins [24]. The cellular receptors used by these viruses to enter cells are different. RSV receptors include heparan sulfate, nucleolin, epithelial growth factor (EGF), and chemokine receptor CX3CR1, which enter cells [25]. In contrast, the SARS-CoV-2 main receptor is an angiotensin II-converting enzyme (ACE2) associated with transmembrane serine protease 2 (TMPRSS2). Complementary receptors for SARS-CoV-2 have been discovered, including CD147, dipeptidyl peptidase 4 (DPP4), and neurophilin (NPRP1) [26] (Figure 1). Both viruses present a similar mutation rate around 10^−3^ nucleotide substitutions per site [27,28].

Previous studies have attempted to explain the differences between SARS-CoV-2 infections in adults and children based on viral entry receptor expression. Lower ACE2 expression in the nasal epithelium has been reported in children compared to adults [29]. However, recent studies have shown that the expression levels of ACE2, Furin protease, TMPRSS2, and Cathepsin (B, L1, V) are similar in the upper airways of children and adults [30]. The expression of ACE2 is low in developing mouse lungs, ranging from embryonic day 18 to postnatal day 64, and is restricted to epithelial cells [31]. In addition, expression levels of ACE2 are low in human lung tissue samples from infants, children, and adults [31]. TRMPRSS2 is lower at earlier murine lung developmental time points but increases with time [31]. Similarly, adult subjects showed higher TMPRSS2 expression than pediatric subjects [31]. Evidence suggests that SARS-CoV-2 replication in primary nasal epithelial cells at the air–liquid interface is lower in children than in adults [32]. The Omicron variant replicates better in pediatric nasal primary cells than the ancestral virus [32]. However, when analyzing nasopharyngeal samples using RNA-seq, the SARS-CoV-2 viral load was found to be similar between children and adults and did not correlate with ACE2 expression [30].

Airway epithelial cells are the primary targets of RSV and SARS-CoV-2 in the nasal mucosa, and provide a physical and immunological barrier to infection. The respiratory epithelium consists of a varied array of cells, each performing distinct functions, and is classified as pseudostratified columnar ciliated. It mainly consists of basal cells and a variety of secretory cells, including goblet cells responsible for mucus generation, club cells, and positive multiciliate cells with coordinated ciliary movement. The cellular composition of the upper respiratory tract is different between children and adults; epithelial cell populations show age dependency; children have more goblet cells and fewer ciliated cells than adults [33]. In the lungs, type 1 pneumocytes (AT1) are specialized alveolar cells that form a lining along the alveolar surface. Its primary function is to facilitate gas exchange in the lungs. Conversely, type 2 pneumocytes (AT2) have a cuboidal shape and are another type of alveolar cell that plays a crucial role in the production and secretion of surfactant proteins [34]. In bronchoscopy biopsy samples, there was no difference in the proportion of cells in the pediatric and adult tracheobronchial epithelia during homeostasis [35]. The balance between pathogen clearance and immune modulation is tightly controlled by the epithelial–immune cell axis during the initial respiratory pathogen infection. The dysregulation of this process can lead to significant tissue damage. The ciliated epithelium serves as a mechanical mechanism for removing debris, allergens, and potential pathogens from the upper respiratory tract, whereas secretory cells produce mucus composed of mucin proteins, cytokines, complement factors, antimicrobial peptides, secretory antibodies, and commensal bacteria lining the airways and mucosal barriers. Mucins, which are diverse glycoprotein complexes, play a crucial role in trapping and preventing viral entry in infected hosts [36].

The defense of the respiratory tract depends on the integrity of epithelial cells and the junctional complex between them. Under normal conditions, the respiratory tract efficiently eliminates external materials without generating excessive inflammatory or immune responses [37,38]. Defense against pathogens in the respiratory tract involves mechanisms in both the upper and lower airways, as well as within the alveolar space. In the upper airways, deposition and elimination occur through coughing, sneezing, and ciliary movement. Macrophages play a pivotal role in removing particles and microorganisms from alveoli [37,38].

Epithelial cells are integral components of the innate immune system, functioning as a physical barrier and producing antimicrobial agents and danger-associated molecular patterns (DAMPs). They also contribute to the specific recognition of antigens through pattern recognition receptors (PRRs) [37,39]. Any disruption in the homeostasis of these systems may contribute to inflammatory diseases of the upper respiratory tract [39].

During RSV infection, ciliated cells are replaced by secretory cells in the nasal epithelium of children [30]. Children infected with SARS-CoV-2 present a higher relative abundance of ciliated cells than children infected with RSV [30]. RSV infection, by replacing ciliated cells with secretory cells, may contribute to excessive mucus production and the occlusion of small airways. Maintaining a balance between secretory and ciliated cells in the lungs is vital for proper lung function [40].

In the lungs, RSV and SARS-CoV-2 infect both type 1 and type 2 pneumocytes [41,42]. However, some studies have suggested that the major target for SARS-CoV-2 is type 2 pneumocytes (AT2) [43,44], whereas RSV targets ciliated cells in the bronchial epithelia and type 1 pneumocytes (AT1) [45]. Type 1 pneumocytes account for 95% of the alveolar surface. Once these cells are infected, the architecture of the alveoli is deregulated, and to replace dying cells with new ones, type 2 pneumocytes differentiate into type I cells, acting as a progenitor cell population for type 1 cells [46]. Pneumocyte injury can occur in all kinds of pulmonary diseases, and this inflammation may result in histological markers such as necrosis, erosion, hyaline membrane, and fibrin exudation with intraluminal granulation tissue formation [47]. RSV infection is related to the pathogenesis of pulmonary fibrosis. This happens because when type 1 pneumocytes are destroyed, type 2 pneumocytes undergo hyperplastic proliferation to cover the missing cells. As this process is too fast, it results in the aberrant replacement and repair of the alveolar surface [48].

A study using a human nose organoid culture model at the air–liquid interface (ALI) showed similar levels of RSV and SARS-CoV-2 virus replication, but RSV replicated on the apical cells, whereas SARS-CoV-2 replicated in both apical and basolateral cells [49]. In addition, SARS-CoV-2 causes more ciliary damage and epithelial thinning than RSV; in contrast, RSV induces more mucus secretion [49]. The tropism of SARS-CoV-2 for basal cells and club cells was described in a study that performed single-cell (sc) RNA sequencing (RNA-seq) of experimentally infected human bronchial epithelial cells (HBECs) in ALI culture; however, replication starts on ciliated cells [50]. In lung tissue, ACE2 and TMPRSS2 are predominantly expressed in transient bronchial secretory cells, as determined by single-cell sequencing [51].

Differences in innate, adaptive, and heterologous immunity, along with differences in endothelial function and coagulation, seem to be the most reasonable mechanisms underlying the significant age-related patterns during SARS-CoV-2 infection.

In children, the innate immune response to SARS-CoV-2 is generally swifter and more robust, particularly in the nasal mucosa, effectively containing the virus at an accelerated rate [51]. Conversely, adults may experience an exaggerated, dysregulated innate response that is less effective, leading to the uncontrolled production of pro-inflammatory cytokines and subsequent tissue damage [52]. However, this more robust response of SARS-CoV-2 in children, which seems to be protective, differs during RSV infection.

Consequently, in this review, we aimed to compare the innate and adaptive immune responses induced by RSV and SARS-CoV-2 to better understand the differences in the pathogenesis of infection in children and to investigate whether it is the immune response that leads to children infected with SARS-CoV-2 not experiencing severe infection, whereas those infected with RSV do.

## 2. General Immunology of Viral Respiratory Infection in the Mucosal Airway

The respiratory tract is the second largest surface area of the body and is in constant contact with antigens and pathogens. Host defense against viruses involves multiple immune components, and the initiation of this response differs depending on the mechanism by which viruses enter, replicate, and spread in the host. Access to target tissues presents numerous obstacles for entry and infection by most viruses, such as the mechanical barriers provided by the mucosal surfaces, demonstrating their high effectiveness in impeding viral entry [53].

Mucosal defense mechanisms are essential for preventing the infection of the respiratory tract by viruses. As the airway mucosa is the major site of gas exchange between the lung and blood, a proper balance between the immune response and tissue function needs to occur. Nasopharynx-associated lymphoid tissue (NALT) plays a role in inducing mucosal immune responses in the upper respiratory tract [37,39,54]. Another associated lymphoid tissue is BALT, localized in the lower respiratory tract, which includes the trachea, bronchi, and branches of bronchi in the lung [36].

In addition to these structural tissues, several immune cells are also associated with an immune response in the airway mucosa. They include dendritic cells, innate lymphoid cells, alveolar macrophages, and tissue-resident lymphocytes. Following contact with the virus, local and circulating cells respond to site insults. Cells present in the mucosa can be interconnected with other mucosal tissues, and this pathway of circulation is described as a common mucosal immune system [55].

The replication of most viruses is restricted to specific target tissues, primarily because of the distribution of viral receptors [53]. After binding to a cellular receptor, viruses can fuse with the cell membrane, enter through endocytosis, and enter the cytoplasm or nucleus via vesicle fusion. Alternatively, they can cross the cell membrane, or cause an endocytic vesicle to rupture upon entering the cytoplasm. Viruses then utilize host cell machinery and specialized proteins encoded by the virus to quickly reproduce within the cell. Once they multiply within the cell, many viruses induce cytolysis to facilitate the release of new infectious. Other viruses are released from infected cells by budding through the cell membrane. However, viruses encounter numerous innate defenses and activate the components of adaptive immunity [53]. The innate immune response is the first line of defense against pathogens, and is activated to block or inhibit initial infection, protect cells against infection, or eliminate virus-infected cells [53]. These responses are orchestrated by different cell types, including neutrophils, monocytes, macrophages, dendritic cells (DCs), and natural killer (NK) cells.

Neutrophils play a crucial role in the immune response against bacterial infections through phagocytosis and the release of antimicrobial substances. Neutrophil subpopulations vary in their capacity to initiate antimicrobial responses, encompassing actions like neutrophil extracellular trap formation, degranulation, and the synthesis of cytokines and chemokines [56]. Nevertheless, their role in viral infections is complex.

The innate response to respiratory viruses is triggered by a key family of receptors known as pattern recognition receptors (PRRs), which recognize pattern-associated molecular patterns (PAMPs) to induce an inflammatory response during infection [57]. PRRs can be represented mainly by toll-like receptors (TLRs), RIG-like receptors (RLRs), NOD-like receptors (NLRs), interferon gene stimulators (STING), and C-type lectin receptors. These molecules signal through various adaptor proteins and activate interferon regulatory factors (IRFs) and nuclear factor-κB (NF-κB). Additionally, this process involves the initiation and activation of the inflammasome, cellular stress, and, in certain cases, cell death. Antiviral innate immunity is characterized by the secretion of cytokines called interferons (IFN), which inhibit viral replication. IFNs are produced and secreted by infected cells and act on uninfected cells to stimulate the production of interferon-stimulated genes (ISGs), which induce an antiviral state through a range of mechanisms [57].

The RIG-I-like receptors (RLRs) RIG-I, MDA5, and LGP2 have a crucial role in detecting RNA virus infections, triggering, and regulating antiviral immunity [58]. RIG-I and MDA5 are cytoplasmic viral RNA sensor proteins, comprising domains, N-terminal caspase activation and recruitment domains (CARDs), RNA helicase, and a C-terminal (CTD). RIG-I recognizes 5′ tri- or diphosphate double-stranded RNA (dsRNA) and prefers relatively short dsRNA, whereas MDA5 binds longer dsRNA [56]. RIG-I and MDA5 use the MAVS adaptor to activate TBK1, which phosphorylates IRF3, inducing type I IFN gene transcription [58].

There are three families of interferons (IFN), type I, type II, and type III, based on sequence homology, receptors, and functional activities. Type II interferons are represented by a single member, IFN-γ. Type I are represented by multiple IFN-α subtypes (13 in humans) and single IFN-β, IFN-ε, IFN-κ, and IFN-ω, all signaling through the IFNAR receptor. Type III interferons are represented by IFN-λ1 (IL-29), IFN-λ2 (IL-28A), IFN-λ3 (IL-28B), and IFN-λ4, and all signal through the IFNLR receptor [59]. Type III interferons are recognized as a primary line of defense at mucosal barriers, effectively combating viral threats without triggering widespread inflammation. In contrast, type I interferons are believed to be produced by the body if the viruses escape this local response. Receptors for type I interferons are expressed in nearly all types of cells, whereas receptors for type III interferons are primarily found in epithelial cells and in a small number of other cell types [59].

Interferons bind to their receptors and initiate their signaling through the JAK/STAT pathway. The result of this signaling is the transcriptional induction of different genes, call ISGs (interferon-stimulated genes). These ISGs are responsible for the expression of direct antiviral effectors or molecules with the potential to regulate IFN signaling and other host responses, directly control viral replication, and they also can present a paracrine function protecting against the spread of viral infection to neighboring cells [60].

Among the ISGs, ISG15 is a ubiquitin-like protein that marks many hundreds of cellular proteins, altering their fate, and conjugation with ISG15 can lower the viral load. In addition, another ISG, the 2′-5′-oligoadenylate synthetase (OAS), plays a crucial role as a key detector of cytosolic double-stranded RNA (dsRNA). Its significance lies in its ability to activate the latent ribonuclease (RNase L), thereby impeding viral replication and establishing an antiviral state, which is pivotal in limiting viral infections [61].

Innate immunity generally serves to slow, rather than stop, viral infection, affording time for the adaptive immune response to initiate [53]. The innate and adaptive immune responses operate in a highly interconnected manner, working collaboratively to generate an effective defense against pathogens. Adaptive immunity is required when the innate immune system’s efforts fall short of eradicating infectious agents. This process is triggered by the delivery of antigens and activated antigen-presenting cells to the draining lymphoid tissues. Different components of the immune response can initiate the adaptive response, which is notably tailored to the specific pathogen responsible for its initiation. In this orchestration, DCs assume a significant role by acting as antigen presenters, stimulating the activation of T and B lymphocytes, crucial players in the adaptive immune response [62].

The adaptive immune system consists of three main cell types: B cells, responsible to produce antibodies; T CD4 cells, which are both helper and effector cells classified into different subtypes (Th1, Th2, Th17, and Treg); and T CD8 cells, with cytotoxic effects towards virus-infected cells. The adaptive immune response is essential for controlling and eliminating viral infections. In addition, adaptive immunity can generate immunological memory, which can quickly eliminate a specific pathogen in a second encounter, preventing subsequent complications [53].

The two primary divisions of adaptive immunity are antibody and T-cell-mediated. Antibodies generally function by binding to free viral particles to block the infection of the host cell. In contrast, T cells primarily function by identifying and eliminating virus-infected cells. Given that viruses typically replicate within cells and many of them spread directly between cells without returning to the extracellular environment, resolving infections relies more on T cell activity than on antibodies. Nonetheless, antiviral antibodies gain significant importance as an added immune defense against subsequent infections. The existence of antibodies at entry points, frequently mucosal sites, is particularly relevant [53].

Unlike the memory triggered by the adaptive immune response, there is also a type of memory associated with innate immunity, which is called immune training. This trained immunity is influenced by past interactions with pathogens and their products and involves various cell types, including myeloid cells, NK cells, and innate lymphoid cells (ILCs) [63]. Trained immunity depends on a changed functional state of innate immune cells, which persists for weeks to months after the initial stimulus is eliminated. A distinctive feature of trained innate immune cells is their ability to create a qualitatively different, and somewhat quantitatively stronger, transcriptional response compared to untrained cells when exposed to pathogens or danger signals [63]. This is made possible by using effector and recognition molecules encoded in the germline, which are different from those involved in conventional immune memory. Trained immunity also exhibits an increased ability to respond to secondary stimuli that are not specific to a particular pathogen. This enhanced response is facilitated by signals that affect transcription factors and induce epigenetic reprogramming, leading to changes in cellular physiology without permanent genetic alterations like mutations or recombination [64].

In the context of infectious diseases, trained immunity can be either beneficial or harmful, depending on the host’s condition, transmission vectors, infecting pathogen, and disease context. Certain viral infections have been shown to reprogram the host’s innate immune response, as seen with RSV. The activation of the thymic stromal lymphopoietin (TSLP) signaling pathway after early-life RSV infections induces chromatin modifications in DCs. This leads to a persistent “trained” phenotype in the lungs, with an activated pathogenic gene program and heightened allergic responses. This demonstrates how RSV-associated trained immunity contributes to disease progression [65]. Also, trained immunity induced by the BCG vaccine has been suggested to induce protection against SARS-CoV-2 infection [66].

## 3. Comparing the Immune Responses to RSV and COVID-19 in Children

### 3.1. Innate Immunity

Neutrophil function is reduced in newborns but increases with age [67]. RSV infection is associated with a significant influx of neutrophils into the airway; however, it can also contribute to both antiviral defense and lung pathology [68,69]. During SARS-CoV-2 infection, neutrophils are linked to the occurrence of interstitial lung modifications in the post-COVID-19 phase [70], and complications such as thrombosis and MIS-C in children [71]. Compared to adults infected with SARS-CoV-2, children have a higher expression of CCL3 and CXCR 1 and 2 in neutrophils in the upper respiratory tract [72]. In addition, compared with older children, infants infected with SARS-CoV-2 have fewer neutrophils in the blood [73]. Despite differences in the roles of these cells, no differences were found in neutrophil levels between children infected with SARS-CoV-2 and RSV [30].

The summary of the innate response in the context of the antiviral response during RSV and SARS-CoV-2 is provided in Table 1.

#### 3.1.1. RIG-I and MDA5

Healthy children have a significantly higher baseline expression level of the genes encoding RIG-I, MDA5, and LGP2 in epithelial cells in the upper respiratory tract than adults [33]. This result suggests an enhanced capacity of the children’s respiratory mucosa to respond to viral infections, which is further supported by the elevated presence of innate immune cells in their upper airways. In contrast, comparing IFN-α production after RIG-I activation in blood mononuclear cells obtained from neonates, young children, and healthy adults, it was observed that both young age and premature birth were linked to reduced RIG-I-dependent IFN-α production [74].

During RSV infection, MDA5 is a known target of viral non-structural protein types 1 and 2 (NS1 and NS2). NS2 inhibits IFN responses stimulated by viral infection by inactivating early-stage interferon signaling binding, and inhibiting the ubiquitination of RIG-I and MDA5 [75]. In RIG-I, NS2 binds via interactions with N-terminal CARDs [76]. The RSV NS1 protein targets TRIM25 to inhibit RIG-I ubiquitination and subsequent antiviral signaling [77]. Silencing MDA5 did not increase RSV transcripts, suggesting that MDA5 might not be essential for controlling virus replication [78]. In contrast, RSV replication in RIG-I-deficient cells is greater than in IFNAR-deficient cells, suggesting that RIG-I is important for viral replication [79]. The importance of RIG-I against RSV infection is also demonstrated by the presence of severe respiratory infections caused by RSV in children with mutations that lead to the loss-of-function of IFIH1, which encodes RIG-I [80]. Similarly, RIG-I activation inhibits RSV replication in mice and ferret models [81]. In addition, mice that lack MAVS, the adaptor of the RIG-I and MDA5 signaling pathways, have higher lung viral titers and delayed viral clearance [82]. MAVS is also essential for T cell responses during secondary RSV infection in mice [83].

As mentioned above, children present a higher baseline level of RIG-I in the respiratory tract than adults, and RIG-I is important to control RSV infection; however, they are highly susceptible to RSV infection. It is possible that viral escape mechanisms that inhibit RIG-I in the respiratory tract are very effective in these populations.

During SARS-CoV-2 infection, RIG-I expression is crucial in the initial defense of lung cells, restricting viral replication in a type I/III interferon-independent manner [84]. SARS-CoV-2 proteins demonstrate the ability to suppress RIG-I in different ways. The NSP7 viral non-structural protein binds to RIG-I and MDA5, preventing the interaction with MAVS, consequently leading to a decrease in the production of type I and type III IFNs [85]. The non-structural viral protein NSP5 reduces the K63-linked ubiquitination of RIG-I to restrict IFN induction [86] and reduces transiently expressed RIG-I and MAVS proteins [87]. Additionally, it cleaves the N-terminal end of RIG-I-N, causing loss of function [87]. ORF9b, an accessory protein of SARS-CoV-2, seems to antagonize RIG-I-MAVS because, in the absence of ORF9b, SARS-CoV-2 RNA was found to activate RIG-I-MAVS signaling [88].

Since SARS-CoV-2 can also inhibit RIG-I signaling, and, consequently, the IFN response, SARS-CoV-2 infection in children has been associated with strong antiviral activity. One possible explanation is that, in contrast to RSV, there is a limited timeframe within which cells can express IFN before SARS-CoV-2 evades the antiviral system [52].

#### 3.1.2. Interferons

In nasopharyngeal samples from children infected with RSV and SARS-CoV-2, the IFN response signature was present in most virus-infected participants, although the expression of secreted type I, type II, or type III IFNs at the transcriptional level was not detected in most patients [30]. This result suggests that the nasal mucosa may not be a major source of secreted IFNs in RSV and SARS-CoV-2 infections. Indeed, RSV is considered a poor interferon inducer in the respiratory tract [89]. Infants infected with RSV in the peripheral blood presented a greater activation of interferon-related genes in samples 4–6 weeks after infection than in the acute phase of the disease, indicating that acute RSV infection in infants might inhibit the interferon response [90]. RSV protein G, more specifically the CX3C motif, can impair type I and III interferon production [91]. RSV F protein can activate EGFR, decreasing the release of IFR1, and, consequently, the production of IFN-λ 1 (IL-29), increasing the viral load and the production of mucus by RSV in airway epithelial cells [92]. In contrast, IFN-λ 1–3 levels have been associated with the severity of respiratory disease caused by RSV, which is related to an increase in respiratory rate during infection [93]. The kinetics of the response might be important for the antiviral effect of interferon. Interferon treatment in RSV-infected patients has been tested [94]. IFN-α1b ameliorates RSV pneumonia symptoms in neonatal infants [95].

Impaired IFN response has been described to be related to severe COVID-19 [93,96,97]. Efficient type I and III responses might be responsible for the control of SARS-CoV-2 infection in children [33]. This population has a preactivated IFN response in epithelial cells and a more robust response in immune cells, affording enhanced protection against the infection in comparison to adults [98].

There is heterogeneity in the IFN response in the respiratory tract during SARS-CoV-2 infection. In mild COVID-19 patients, the presence of IFN-λ1 and IFN-λ3, but not IFN-λ2 or IFN-I, was related to the induction of ISGs, responsible for containing SARS-CoV-2 infection in a more efficient way [99]. The results of controlled phase III clinical trials (WHO SOLIDARITY, ACTT-3, and SPRINTER) did not demonstrate a significant therapeutic effect of type I interferons in COVID-19 [100]. Conversely, in a randomized controlled phase III trial (TOGETHER), a significant decrease in the hospitalization rate was found when patients were treated with type III interferon [101].

#### 3.1.3. ISGs

Previous studies have shown that ISG15 exhibits an anti-RSV effect, which exerts its antiviral activity via the protein ISGylation. For this antiviral activity to occur, the presence of high levels of ISG15 in cells prior to infection is required. ISG15 is also upregulated in pseudostratified respiratory epithelium and in nasopharyngeal samples from RSV-infected children, corroborating the antiviral role of the molecule [102]. To evade the antiviral response, some viruses try to interfere with ISG15 function. SARS-CoV-2 protein papain-like proteases (PL1pro) which function is to cleave viral polyproteins can also after translation conjugate with poly-ubiquitin and protective ISG15 [103]. A large-scale study of ISG function during SARS-CoV-2 infections found that 65 ISGs play a role in the control of virus replication. These processes encompass endocytosis, nucleotide biosynthesis, and sphingolipid metabolism, demonstrating the diverse functional impact of these ISGs. Bone marrow stromal antigen 2 (BST2, also known as CD317 or tetherin) was found to impair viral egress and be antagonized by the SARS-CoV-2 accessory protein Orf7a [104].

The family of enzymes known as 2′-5′-oligoadenylate synthetase (OAS) play a crucial role as key detectors of cytosolic double-stranded RNA (dsRNA). Their significance lies in their ability to activate the latent ribonuclease (RNase L), thereby impeding viral replication and establishing an antiviral state, which is pivotal in limiting viral infections [61]. On one hand, regarding RSV infections, interferon-related genes such as RSAD2, LAMP3, IFI44L, and OAS1 are significantly highly expressed in individuals infected with RSV who present symptoms [90]. On the other hand, children who have presented severe cases of COVID-19, such as multisystem inflammatory syndrome in children (MIS-C), have shown deficiencies in OAS1, OAS2, and RNASE, potentially affecting the antiviral responses against the affected cells [105]. These findings reveal a contrasting role compared to severe SARS-CoV-2 infections in children. While deficiencies in OAS1 seem to influence the antiviral response to SARS-CoV-2 in children, the same gene appears to play a relevant role in RSV severity, suggesting a distinct connection between these two viruses and their clinical manifestations.

## 4. Adaptive Immune Response

Adults and children have differences in their immune responses to viral infections. The function of innate immunity is equal to adult levels in early childhood, while B and T cell functions continue to develop during childhood. However, the regulatory function of T cells decreases with aging, which is much more significant in pediatric patients [106].

During RSV infection, a robust memory B cell response is induced in the adenoids of young children [107]. In children, nasal antibody titers correlate with protection against virus infection [108]. Infant immune responses exhibit a bias towards recognizing a limited set of antigenic sites, which differ from the ones predominantly targeted by adult immune systems [109]. Disease severity in infants hospitalized with RSV has been demonstrated to correspond with the level of immunoregulatory IL-10-expressing B cells, which diminish protective Th1 responses, in the nasopharyngeal samples [110]. Studies suggest that in infants, excess type 2 and/or deficient type 1 immune responses are associated with the pathogenesis of RSV bronchiolitis [111].

The Th2 response plays a significant role in RSV infections. Th2 response is a type of immune response characterized by the activation of type 2 helper T lymphocytes, which produce specific cytokines that have diverse effects on the immune response and inflammation. It is involved in generating antibody responses, particularly IgE, and eosinophil responses. Moreover, it might counteract and limit Th1-mediated inflammation, being linked to more severe disease presentations. In the context of RSV infections, the Th2 response is particularly relevant because it could be associated with some of the observed complications, especially in children, such as the development of bronchiolitis, heightened inflammation, and an imbalance with the Th1 response [112].

Distinct T cell profiles were observed to be related to severe respiratory syncytial virus (RSV) disease. CD8+ T cells expressing IFN-γ (Tc1) and IL-17 (Tc17) and Th17 were associated with a shorter hospitalization, in contrast, CD8+ T cells expressing IL-4 (Tc2) were associated with pathology [113]. RSV infection during infancy has been found to have long-term effects on memory T cell responses. RSV-infected infants during their first year of life exhibit diminished antiviral memory T cell responses to RSV by the age of 2–3 years in comparison to children who remained uninfected by RSV infection in their first year of life [114].

RSV also has several ways to escape or evade the host’s immune response, such as mutations in the F or G protein, as observed in children treated with palivizumab, conferring resistance to it and another monoclonal antibody, motavizumab [20]. These mutations also make it harder for the immune system to recognize the virus. Moreover, RSV can trigger an unbalanced immune response [90,115], leading to inflammation and tissue damage [116]. This not only impairs the immune system’s ability to combat the virus but can also result in more severe symptoms.

As previously described, children usually do not have a severe clinical presentation in response to the SARS-CoV-2 virus. According to a study, among patients with mild COVID-19, children have a lower likelihood of seroconversion than adults, despite similar viral loads [117]. When comparing children of different ages (2–15 years), they found similar levels of antibodies [118].

A well-described marker of severity in COVID-19 is T cell lymphopenia in peripheral blood and in the respiratory tract mucosa, both in children and adults. However, when comparing the levels of these cells in pediatric patients and adults, children demonstrated a higher absolute T lymphocyte count [119]. Nasal samples from adults with asymptomatic to mild COVID-19 presented upregulated T cell activation and TCR signaling compared to children [30].

Unlike adults, children do not develop robust CD4 T cell immunity against the virus. Therefore, in the case of reinfection, the immune system does not recognize the virus and responds as if it was the first encounter [120]. Children develop early N-specific cytotoxic T cell responses, which rapidly expand and establish dominance in their immune memory against the virus [121]. Additionally, there is evidence that the adaptive immune response persists in adenoids and tonsils after SARS-CoV-2 infection [122]. The convalescent children presented an expansion of the germinal center in the pharyngeal lymphoid tissue and T cells producing IFN-γ [122].

Some studies have suggested that children may benefit from pre-existing immunity to other coronaviruses. However, it has been observed that antibody levels are similar in both children and adults [123]. This hypothesis emerges from research indicating that the host can prevent infection with CoV strains through neutralizing antibodies [124], which could also mitigate SARS-CoV-2 infection. Furthermore, inhibiting the interaction between the RBD protein and ACE2 has been observed to be a potentially useful approach for treating SARS-CoV-2 infection [125].

However, SARS-CoV-2 has developed various strategies to elude the host’s immune response, mainly through mutations that have led to multiple variants. SARS-CoV-2 exhibits notable changes in the N and S proteins, along with accessory proteins, impacting its infectivity, and how easily it spreads and recognizes antibodies [126].

A comparative study analyzing the T cell response of children with RSV or SARS-CoV-2 pneumonia showed an increase in the CD8 T cell population in children infected with SARS-CoV-2 in peripheral blood [127]. Similar to RSV infection, children with acute SARS-CoV-2 infection also present an increased Th2 response compared to healthy control children [128].

## 5. Conclusions

RSV and SARS-CoV-2 viruses present intrinsic differences in the replication cycle, cell susceptibility, and host response evasion mechanisms. Children with COVID-19 often experience mild or asymptomatic cases, whereas RSV can cause a range of respiratory illnesses, including bronchiolitis and pneumonia. The differences in susceptibility between RSV and SARS-CoV-2 infections can be influenced by individual factors, with no single response applicable to all situations. A summary of the general immune response during RSV and SARS-CoV-2 infection is described in Figure 2. In comparison to RSV, children might exhibit a more robust immune response against SARS-CoV-2; however, the specific factors driving these distinct responses remain unclear.

## Figures and Tables

**Figure 1 biology-12-01223-f001:**
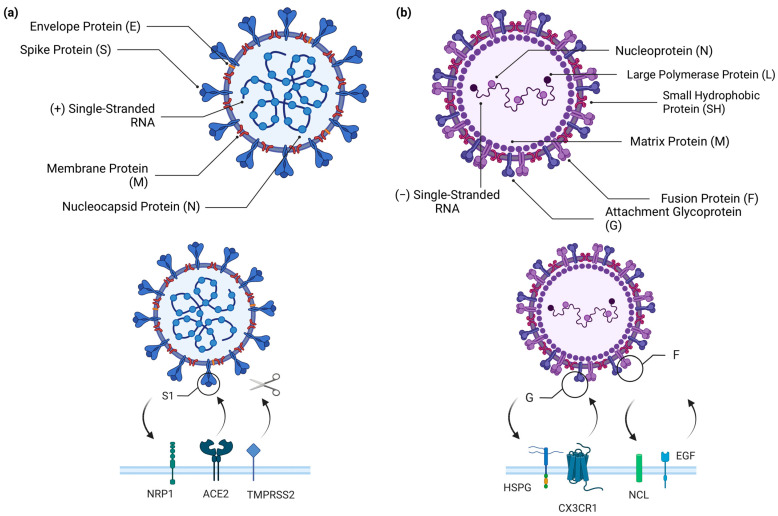
Structures of SARS-CoV-2 and RSV and the host receptors to which they bind. (**a**) Schematic representation of the composition of SARS-CoV-2 virion, which consists of a spike protein (S), envelope protein (E), membrane protein (M), nucleocapsid protein (N), and positive-sense single-stranded RNA (+ssRNA). To enter the host cell, SARS-CoV-2 primarily binds to the angiotensin-converting enzyme 2 (ACE-2) receptor, using the S1 portion of the spike protein. After the binding between the spike protein and ACE-2, the transmembrane serine protease 2 (TMPRSS2) cleaves the spike protein to allow fusion of the viral and host cell surfaces. SARS-CoV-2 also has an alternative entry pathway through neuropilin-1 (NRP1), which also binds to the spike protein. (**b**) Schematic representation of the composition of the RSV virion, which presents an attachment glycoprotein (G), fusion protein (F), small hydrophobic protein (SH), matrix protein (M), nucleoprotein (N), polymerase protein (L), and a negative-sense single-stranded non-segmented RNA (−ssRNA). RSV can bind to various receptors, including CX3CR1 and heparan sulfate proteoglycan (HSPG), which mediate virus attachment and bind to the G protein of the virus, and nucleolin (NCL) and the epidermal growth factor receptor (EGFR) which mediate virus internalization and promote fusion of the RSV virus, respectively. Created with BioRender.com.

**Figure 2 biology-12-01223-f002:**
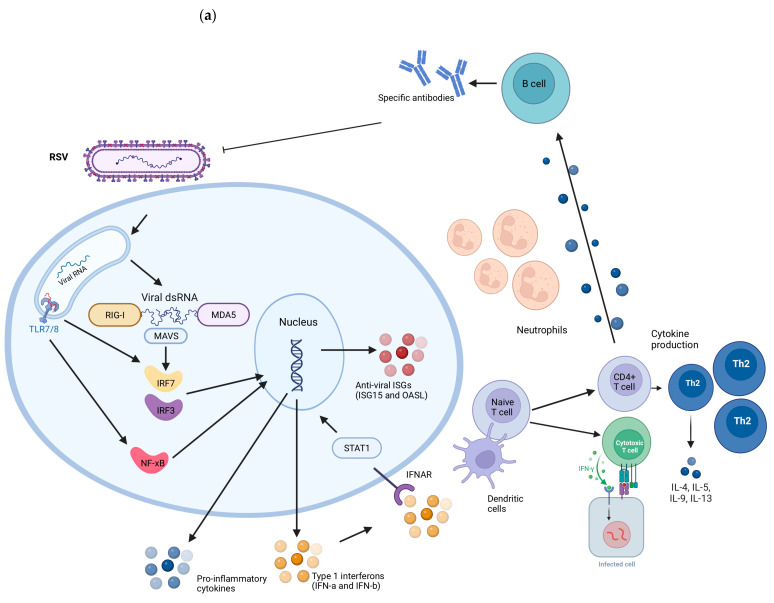
Representation of the immune response induced during RSV (**a**) and SARS-CoV-2 (**b**) infections. RIG-I and MDA5 receptors detect the presence of viral genetic material. When activated, they signal through the Mitochondrial Antiviral Signaling Protein (MAVS), activating transcription factors IRF3 and IRF7. They migrate to the cell nucleus, where they promote the transcription of genes responsible for the production of type I interferons (IFNs), such as interferon alpha (IFN-α) and interferon beta (IFN-β). IFNs activate STAT1 and induce the expression of Interferon-Stimulated Genes (ISGs) inside cells, which have antiviral functions, aiding in the containment of infection. Extracellular, dendritic cells capture viral antigens and present fragments to CD4 and CD8 T lymphocytes. CD4+ T cells are activated into specific subpopulations, such as type 2 helper T cells (Th2), which secrete inflammatory cytokines. Th2 cells play a particularly important role in the response to RSV as they regulate the production of mucus and airway secretions. B cells activated by T cells and cytokines produce specific antibodies against these viruses. CD8 T cells induce apoptosis in infected cells and are increased in children infected with SARS-CoV-2 compared to RSV-infected children. Created with BioRender.com.

**Table 1 biology-12-01223-t001:** Innate immune response to RSV and SARS-CoV-2 infections.

	RSV Infection	SARS-CoV-2 Infection
IFN-λ1 and IFN-λ3	IFN-λ1 and IFN-λ3 are associated with the severity of respiratory disease caused by RSV	IFN-λ1 and IFN-λ3 are associated with the induction of ISGs, efficiently containing SARS-CoV-2
RIG-I and MDA5	NS1 protein inhibits ubiquitination of RIG-I and MDA5	NSP7, NSP5, and ORF9b interact with RIG-I and MDA5, preventing the interaction with MAVS
IFN-I	IFN-I response is impaired during RSV infection	IFN-I response is impaired during SARS-CoV-2 infection
IFN-III	IFN-III response is impaired during RSV infection	IFN-III response is impaired during SARS-CoV-2 infection
OAS1	OAS1 is significantly expressed in individuals with symptomatic RSV	OAS1 is deficient in children who have presented severe cases of COVID-19
ISG15	ISG15 is upregulated during RSV infection	SARS-CoV-2 tries to interfere with ISG15 function

## Data Availability

Not applicable.

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
