# Peer review of "Exploring the Immune Response against RSV and SARS-CoV-2 Infection in Children"

_biology, 2023, doi:10.3390/biology12091223_

Round 1

Reviewer 1 Report

Comments to Authors:

Although English language is fine and does not require editor separately, however few edits have been suggested for incorporation. 

Author Response

Reviewer 1’s Comments to Authors:

Comparing the immune response to Respiratory Syncytial Virus (RSV) and SARS-CoV-2 in children can indeed be a potentially interesting and meaningful comparison. Both viruses primarily affect the respiratory system, and they can cause severe respiratory illnesses in certain populations, including children. Understanding how the immune response differs between these two viruses in pediatric populations could provide valuable insights for several reasons. I find that this is a well-executed review article comparing the immune responses to RSV and SARS-CoV-2 in children and can hold significant scientific and clinical value. It has the potential to contribute to our understanding of these viral infections, inform public health strategies, and guide future research efforts in managing and preventing respiratory illnesses in pediatric populations.

Our response: We thank the review positive comments and we addressed all the suggestions to improve the manuscript.

I have the following queries and suggestions:

Reviewer comment 1. The abstract and main body highlights that immune response amongst children is being compared (RSV vs SARS-CoV2). Why should the title miss this- amongst children?

Our response: We have now added children in the title.

Reviewer comment 2. What formed the basis of the comparison of immune responses amongst children from two respiratory viruses with drastically distinct characteristics?

The second question may further be answered after going through the following recent literature:

https://www.ncbi.nlm.nih.gov/pmc/articles/PMC9538042/

https://www.thelancet.com/journals/lancet/article/PIIS0140-6736(2200383-X/fulltext

https://www.nature.com/articles/s41577-022-00764-7 This one may also be useful for conclusion elaboration and the next question below:

Our response: The basis of the comparison of immune response was due to the fact that

children infected with RSV develop severe disease, while in cases of SARS-CoV-2 infection in children, the disease tends to be less severe compared to adults. We discussed in the manuscript, line 61, the literature suggested by the Reviewer. We added a new paragraph about the RSV and SARS-CoV-2 co-infection and RSV seasonality during pandemic, which complement the distinct characteristics of these two viruses. 

Reviewer comment 3. Can the viral escape mechanisms from the immune system be more elaborated in context to RSV and SARS-CoV and/or otherwise if the other previous knowledge from some specificity switching may relate to it in other CoV strains?

For this 3rd query please read and accordingly include the below citations if appropriate and where relevant:

Reguera, Juan, Cesar Santiago, Gaurav Mudgal, Desiderio Ordono, Luis Enjuanes, and Jose M. Casasnovas. "Structural bases of coronavirus attachment to host aminopeptidase N and its inhibition by neutralizing antibodies." (2012): e1002859.

Dubey, A., Choudhary, S., Kumar, P., & Tomar, S. (2022). Emerging SARS-CoV-2 variants: genetic variability and clinical implications. Current microbiology, 79, 1-18.

Reguera, J., Mudgal, G., Santiago, C., & Casasnovas, J. M. (2014). A structural view of coronavirus–receptor interactions. Virus research, 194, 3-15.

Duan, L., Zheng, Q., Zhang, H., Niu, Y., Lou, Y., & Wang, H. (2020). The SARS-CoV-2 spike glycoprotein biosynthesis, structure, function, and antigenicity: implications for the design of spike-based vaccine immunogens. Frontiers in immunology, 11, 576622.

Our response: We thank the Reviewer for the comment and we try to improve the discussion of this topic thought the manuscript. We found the both of virus can evade the immune response, including the antivirus response induced by interferons, as well as, the antibody neutralization. Probably when the host is children RSV has a more elaborate mechanism of evasion leading to a more severe disease. We also added the literature suggested by the Reviewer into the manuscript.

Review comment 4: The review could benefit from a compelling figure highlighting the differences in innate and adaptive immune responses from RSV and SARS-CoV-2. It may potentially draw key responses and dynamics at the respiratory organs/epithelia and/or a proposed cascade of events that lead to these responses in these different viral infection scenarios. In the conclusions section, you mentioned, “Children may have a more protective 363 immune response against SARS-CoV-2 compared to RSV, especially represented by the innate response; however, the specific factors driving these distinct responses remain unclear”. The compelling figure may benefit from highlighted checkpoints of the unknown driving factors, as well.

Our response: We have now added a new Figure 2 summarizing the immune response.

Review comment 5: Other minor edits and suggestions:

See also comments in the comments’ pane on the right where marked.

Line 54: remove – ‘, has increased’

Line 58: approximately 3 in 10.000?? OR approximately 3 in 10,000 ?

Line 65-66: separate the RSV portion and/or merge it in next sentence in line 66 Both viruses are enveloped; however, they have genomes of different sizes and polarities. RSV have a genome of 15kb of negative-sense single-stranded RNA that encodes 11 proteins [19],……. In the above sentence, RSV is being talked about. Please add “RSV”.

Also, mention the polarity of SARS-CoV-2 in the next line.

Line 72: change SARS-Cov-2 to SARS-CoV-2

Line 102: cells

Line 128: remove repetition- expression levels of expression of ACE2 are low in

Line 135: replace ‘was similar’ with ‘was found similar’

Line 146-148 Please elaborate more on why this comparison is noteworthy. Did the rise in RSV cases follow after COVID-19 or if there are other cues that can be exploited as a basis to warrant a review of comparison between these two different viruses in recent times?

Line 176: The summary of the innate response regarding antiviral response during RSV and SARS-CoV-2 is in Table

- Please adjust “regarding” to “in the context of”.

- Please adjust “in” to “provided in”.

In table 1. Please elaborate more than leaving the response ‘impair’. You mean infection is impared? Or IFN-I and III? clarify. Similarly, clarify ‘Interacts with RIG-I and MDA5 preventing the interaction with MAVS’. What interacts with them? See other cells in table as well to clarify.

Line 230: correct SARS-COV2

Line 237-242: cite the statements with appropriate references.

Line 247: do not change the paragraph. Context is continued, this so.

Our response: We corrected all the suggestions

Reviewer 2 Report

In its current state it is not acceptable for publication, as it is confusing and while titled “Differences between—“, it confusingly concludes that “differences remain to be determined”

Some points:

(i) This is a strange paper - it is a list of molecular observations in no particular order and in a disconnected way. Assuming this is for the non-initiated, there is a need to identify “big picture airway immunology (including the role of T cell delivery via the Common Mucosal System”, the importance of “learned innate immunity”, the whole question and data re mucosal immune response, the “connect” between innate and adaptive immunity etc. There must be diagrams illustrating these points, in real time.
(ii) There is a mix of basic explanations, and assumptions the reader actually understands what you are writing. Which I doubt they would.
(iii)I would suggest a complete re-write into something of clear meaning and value. Take just Covid, develop more clearly - first the bigger question as to how airway protection would be expected to occur, then how it does or doesn’t in Covid. Then contrast with other viral infections eg flu; RSV etc, and then show how outcome may be dependent of mechanisms, and how they vary between children and adults.

This could be a very useful paper, but needs a complete re-think.

-

Author Response

Reviewer 2’s Comments to Authors:

Reviewer comment 1 In its current state it is not acceptable for publication, as it is confusing and while titled “Differences between—“, it confusingly concludes that “differences remain to be determined”

Our response: We thank the Reviewer for the comment, we have now change the title to be more appropriated.

Reviewer comment 2: Some points: (i) This is a strange paper - it is a list of molecular observations in no particular order and in a disconnected way. Assuming this is for the non-initiated, there is a need to identify “big picture airway immunology (including the role of T cell delivery via the Common Mucosal System”, the importance of “learned innate immunity”, the whole question and data re mucosal immune response, the “connect” between innate and adaptive immunity etc. There must be diagrams illustrating these points, in real time.

(ii) There is a mix of basic explanations, and assumptions the reader actually understands what you are writing. Which I doubt they would.

(iii)I would suggest a complete re-write into something of clear meaning and value. Take just Covid, develop more clearly - first the bigger question as to how airway protection would be expected to occur, then how it does or doesn’t in Covid. Then contrast with other viral infections eg flu; RSV etc, and then show how outcome may be dependent of mechanisms, and how they vary between children and adults. This could be a very useful paper, but needs a complete re-think.

Our response: We thank for the suggestions. We have change the manuscript format in order to include more general information about the immune response against respiratory virus in the airways to make clear to readers. In the final section of the manuscript we focus on the differences between RSV and SARS in children. We added a figure to help summarized the immune response of RSV and SARS-CoV-2 in general.
